# The efficacy of the fixed combination of latanoprost and timolol versus other fixed combinations for primary open-angle glaucoma and ocular hypertension: A systematic review and meta-analysis

**Yi Xing, Lijuan Zhu, Ke Zhang, Shaohua Huang**[¤]*

Department of Ophthalmology, The First Affiliated Hospital of Zhengzhou University, Zhengzhou, People's Republic of China

¤ Current address: Department of Ophthalmology, The First Affiliated Hospital of Zhengzhou University, Zhengzhou, People's Republic of China
* medhsh@163.com

**Data Availability Statement:** All relevant data are within the manuscript and its Supporting Information files.

## Abstract

### Background

Fixed-combination (FC) therapy is used in primary open-angle glaucoma (POAG) and ocular hypertension (OHT) patients who require more than one medication to reach their target intraocular pressure (IOP). Currently, there are several FC therapies available for the treatment of glaucoma. The FC of latanoprost/timolol (LTFC) is a commonly used FC. Here, we conducted systematic review to compare the IOP-lowering effects of LTFC with other FCs for patients with POAG and OHT.

### Materials and methods

We searched PubMed, EMBASE, the Cochrane Library, and Web of Science for randomized-controlled clinical trials and cross-over studies. The outcomes were mean IOP and IOP fluctuation after one month of treatment. Meta-analysis was carried out using RevMan (version 5.1) software. After conducting meta-analyses, we rated the quality of each meta-analysis as high, moderate, low, or very low using the "GRADE" system.

### Results

We included 16 trials in this meta-analysis. Moderate-quality meta-analysis showed that LTFC had a comparable mean IOP to that of a fixed combination of travoprost and timolol (TTFC) [mean difference (MD): 0.07 mmHg] and a fixed combination of dorzolamide and timolol (DTFC) [MD: −0.31 mmHg], and it also had a comparable IOP-fluctuation effect compared to that of TTFC [MD: 0.13 mm Hg] and DTFC [MD: 0.25 mmHg]. Compared to the fixed combination of bimatoprost and timolol (BiTFC), moderate-quality evidence showed a higher mean IOP in the LTFC group [MD 0.76 mmHg], whereas low-quality meta-analysis showed higher IOP fluctuation [MD 1.09 mmHg] in the LTFC group.

**Funding:** The authors received no specific funding for this work.

**Competing interests:** The authors have declared that no competing interests exist.

## Conclusions

LTFC is as effective as TTFC and DTFC, but worse than BiTFC in controlling mean IOP and IOP fluctuation for POAG or OHT patients. The quality of our meta-analyses was assessed as moderate, with the exception of one low-quality analysis that compared the IOP fluctuation of LTFC and BiTFC.

## Introduction

For patients with primary open-angle glaucoma (POAG) and ocular hypertension (OHT) who require more than one medication to reach their target intraocular pressure (IOP), fixed-combination therapy offers several advantages over concomitant therapy. Fixed-combination therapy is the use of two or more medications that are combined in fixed concentrations into one bottle. Studies have found that fewer bottles and daily drops lead to improved compliance [1, 2]. Fixed-combination therapy also eliminates potential washout effects. Currently, there are several combination agents available for the treatment of glaucoma such as fixed combinations of pilocarpine/timolol, brinzolamide/timolol, dorzolamide/timolol (DTFC), brimonidine/timolol (BTFC), brimonidine/brinzolamide, bimatoprost/timolol(BiTFC), travoprost/timolol (TTFC), tafluprost/timolol (TaTFC), or latanoprost/timolol (LTFC). Most of these agents combine timolol with other hypo-tension medications. Timolol maleate is a non-selective beta-blocker that lowers IOP by reducing the production of fluid; however, it has little activity during night-time hours because aqueous fluid production by the ciliary body is reduced due to natural circadian factors [3]. The prostaglandin analogues work by allowing more fluid to flow out of the eye through the uveoscleral pathway and are therefore less affected by circadian variations in aqueous production. Prostaglandin analogues (PGAs) are considered initial medical therapies for lowering IOP in patients with glaucoma, as they are highly efficacious, well-tolerated, and administered once daily; they are also relatively safe [4]. The fixed combination of 0.005% latanoprost and 0.5% timolol is a commonly used timolol/PGA combination. At present, there are some systematic reviews that have compared the IOP-lowering efficacy of LTFC with that of other fixed combinations [5, 6, 7, 8], but there is a lack of systematic reviews on IOP fluctuation. Moreover, the available meta-analyses lack quality assessments. In the present study, we conducted a meta-analysis of randomized-controlled clinical trials (RCTs) to compare the mean IOP and IOP fluctuation after one month of treatment to identify whether LTFC is as effective as other fixed combinations for patients with POAG and OHT. Additionally, we conducted assessments to assess the quality of these meta-analyses.

## Methods

This systematic review and meta-analysis were performed according to the guidelines outlined in the Cochrane Handbook for Systematic Reviews of Interventions [9]. A formal protocol for this review can be found at dx.doi.org/10.17504/protocols.io.bbsfinbn and a checklist of the PRISMA assessment for this systematic review is given in the S1 File.

### Criteria for considering studies for review

We included all randomized-controlled clinical trials (RCTs) and cross-over studies that compared the efficacy of LTFC with other anti-glaucoma fixed combinations for POAG and OHT patients. The treatment in the intervention group was topical administration of LTFC, which

was administrated via one drop once daily in the morning or evening. The treatments in the control groups were topical uses of other kinds of timolol/PGA combination agents or other clinically available fixed combinations. The outcomes for this review were the mean IOP and IOP fluctuation. If a study only reported the change from baseline, we included change-from-baseline results in the meta-analysis along with endpoint data. The standard time point of measurement was one month; if the trial did not report the IOP at one month, we used the closest time point instead.

## Search methods for identification of studies

We searched the Cochrane Central Register of Controlled Trials (CENTRAL) in The Cochrane Library, MEDLINE (PubMed), EMBASE, and Web of Science. The keywords for the medication were 'timolol', 'latanoprost' and 'fixed drug combinations'. The keywords for the disease were 'primary open-angle glaucoma' and 'ocular hypertension'. The limit for the study was that it had to be a randomized controlled trial. We have attached our full literature search strategy as a supplementary file (S2 File). There were no language or date restrictions; the last search was completed in April 2019. Lastly, we searched the reference lists of identified trial reports to find additional studies.

## Data collection and analysis

After removing duplicate reports, two authors checked the titles and abstracts to exclude any reports that did not meet the inclusion criteria. Then, they independently determined the eligibility of each study by examining the full text and collected data according to a customized form. In order to assess the internal validity of the included studies, the Cochrane Collaborations tool was used to assess the risk of bias. Since only objective results were analyzed in this review, the risk of bias in each study originated mainly from the following five domains: random sequence generation, blinding of participants and personnel, incomplete outcome data, baseline imbalance, and studies that were terminated prematurely. We assigned an assessment of 'low risk of bias', 'high risk of bias' or 'unclear risk of bias' for each domain. We summarized the assessment of these five key domains to analyze the limitation of each study and used the limitation of the study as one of the reasons for assigning a lower quality rating for the meta-analysis.

Next, we summarized the results across studies with mean differences (MD) and 95% confidence intervals for continuous data. We identified statistical heterogeneity with $I^2$ and chi-squared tests. We used sensitivity analysis to evaluate the impact of methodological variables and study risk-of-bias variables on the results of each meta-analysis.

We also rated the quality of each meta-analysis using the "Grades of Recommendation, Assessment, Development, and Evaluation" (GRADE) system [10]. In this system, meta-analyses of RCTs started as a high quality of evidence. Five factors (study limitations, imprecision, inconsistency of results, indirectness of evidence, and publication bias likely) downgraded the rating for the quality of evidence. In addition, two factors (large magnitude, dose response gradient) rased the rating for the quality of evidence. Ultimately, the quality of evidence for each meta-analysis corresponded to one of four categories: high, moderate, low, or very low.

## Results

### Search for relevant studies

Our literature search revealed 2,155 articles (Fig 1). After the removal of duplicate studies, we obtained a total of 1,488 articles. After screening the titles and abstracts, we excluded 1,469

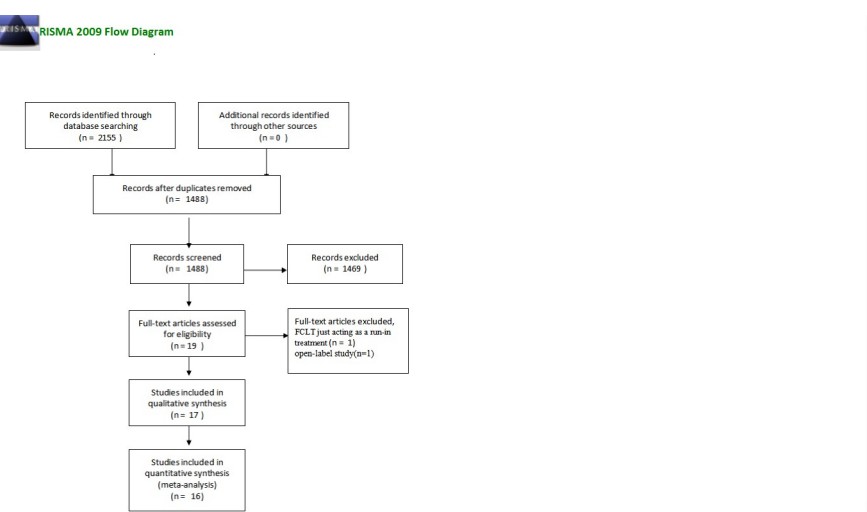

**Fig 1. Flow diagram of the study selection as per the PRISMA statement.**

articles from further assessment. In the end, we obtained the full text of 19 relevant articles. Screening the references of these 19 articles did not reveal any further relevant articles. We then screened each of these 19 articles for their content and methodological qualities. Based on our thorough screening, we excluded one article because LTFC was used only as a run-in treatment before the randomization [11] and another article because it is an open-label study [12]. As a result, a total of 17 articles describing 17 clinical trials that addressed LTFC versus other anti-glaucoma fixed combinations for POAG and OHT patients were included in the systematic review, and 16 of these articles were included in the meta-analysis.

## Characteristics of included studies

The characteristics of these 17 clinical trials are described in Table 1. Among these trials, 8 studies were cross-over studies and nine studies were parallel-arm studies. There were three multi-centre studies that each had more than 100 participants in each group. Thus, this systematic review included a total of 1,566 patients with an average age of 64.6 years, ranging from 50.0 to 73.1 years old. In most clinical trials, the percentage of patients with POAG or OHT exceeded 80% [13, 14, 15, 16, 17, 18, 19, 20, 21, 22].

In the control group, there were four types of anti-glaucoma fixed combinations: 1) TTFC; 2) BiTFC; 3) DTFC; and 4) BTFC. The characteristics of eye-drop administrations and IOP measurements are summarized in Table 2. Only five clinical trials measured and reported 24 hours of IOP curves; the remainder of the clinical trials reported diurnal IOP records. All but one of these studies reported IOP results over one month of treatment.

## Risk of bias in included studies

The risk of bias in the included studies is shown in Table 3. Approximately half of the studies had adequate randomization, but the randomization for the remaining studies was unclear. Eight studies were assessed as having a high risk of performance bias since there was no blinding of patients. Five cross-over studies did not lose any patients. The percentage of patient loss during follow-up was lower than 10% in seven parallel group studies. Six studies in which 'per protocol' analyses were reported were rated as having a high risk of attrition bias. Seven studies did not have a pre-specified sample size; these studies had a high risk of bias due to premature

**Table 1. Characteristics of the 17 included studies.**

| First author/Year/Reference | Study Design | No. of Centers | Treatment | No. of Patients | Mean Age (years) | No. Males | No. Females | POAG or OHT (%) | Funding source |
|---|---|---|---|---|---|---|---|---|---|
| **Pachimkul 2011** [15] | 3P | 1 | LTFC vs TTFC | 10/12 | 50/63.92 | 6/6 | 4/6 | 100/100 | No |
| **Topouzis 2007** [20] | 2P | 41 | LTFC vs TTFC | 200/207 | 64.9/64.8 | 72/64 | 92/104 | 90/88.7 | Alcon. |
| **Konstas 2014** [23] | C | 1 | LTFC vs TTFC | 42 | 65.3 | 20 | 22 | 55 | Alcon, Allergan. |
| **Rigollet 2011** [16] | 3P | 1 | LTFC vs TTFC vs BiTFC | 44/48/49 | 68 | 41 | 87 | 100 | No |
| **Guven Yilmaz 2018** [13] | 3P | 1 | LTFC vs TTFC vs BiTFC | 14/18/18 | 60.4/67/63.2 | 7/8/9 | 7/10/9 | 100 | No |
| **Centofanti 2009** [24] | 2P | 3 | LTFC vs BiTFC | 35/47 | 65.6/64.1 | 16/24 | 19/23 | Unclear | Allergan. |
| **Martinez 2007b** [20] | 2P | 1 | LTFC vs BiTFC | 18/18 | 71.2/73.1 | 10/9 | 8/9 | 50/44 | No |
| **Martinez 2009** [20] | C | 1 | LTFC vs BiTFC | 54 | 71.4 | 28 | 26 | 44 | No |
| **Cvenkel 2008** [19] | C | 1 | LTFC vs DTFC | 32 | 61/68 | 7 | 25 | 100 | Merck. |
| **Eren 2012** [14] | C | 1 | LTFC vs DTFC | 33 | 58.24 | 19 | 14 | 100 | No |
| **Januleviciene 2009** [18] | 2P | 1 | LTFC vs DTFC | 15/15 | 58.13 | 5 | 25 | 100 | Merck. |
| **Konstas 2004** [21] | C | 2 | LTFC vs DTFC | 33 | 64.57 | 13 | 20 | 91 | Unclear |
| **Konstas 2008** [26] | 3C | 1 | LTFC vs DTFC | 31 | 62.8 | 16 | 15 | 58 | Merck. |
| **Martinez 2007a** [27] | C | 1 | LTFC vs DTFC | 32 | 68.9 | 17 | 15 | 59 | No |
| **Miglior2010** [17] | 2P | 25 | LTFC vs DTFC | 135/135 | 65.8/66.6 | 67/54 | 68/81 | 92/91 | No |
| **Shin 2004** [22] | 2P | 30 | LTFC vs DTFC | 125/128 | 64.0/63.0 | 58/54 | 67/72 | 100/100 | Pharmacia. |
| **Hommer 2012** [28] | C | Unclear | LTFC vs BTFC | 18 | 64 | 8 | 10 | 100 | Allergan. |

P, parallel group; C, cross-over; LTFC, 0.005% Latanoprost/0.5% timolol; DTFC, 2.0% Dorzolamide/0.5% timolol; BTFC, 0.2% Brimonidine/0.5% timolol; TTFC,0.004% Travoprost/0.5% timolol; TaTFC, 0.0015% Tafluprost /0.5% timolol; BiTFC, 0.03% Bimatoprost/0.5% timolol; M, morning; E, evening; POAG, primary open-angle glaucoma; OHT, ocular hypertension.

termination. According to the judgments of these key domains, we summarized the risk of bias for each study as follows: Cvenkel (2008) and Eren (2012) = low risk of bias, and Miglior (2010) and Guven Yilmaz (2018) = high risk of bias, while all other studies had a moderate risk of bias.

## Effects of interventions

We conducted meta-analyses to compare the effect of LTFC with other FCs on IOP from a total of 16 RCTs and/or cross-over studies. Among them, three meta-analyses were performed on the mean IOP, and three meta-analyses were performed on the fluctuation of IOP. The results are shown in Table 4 and the quality assessment of these meta-analyses are shown in Table 5. In these meta-analyses, we did not find any factors that could increase the rating for the quality of evidence.

Only one study comparing LTFC to 0.2% Brimonidine/0.5% timolol (BTFC) met the criteria, so we included that study in the system review but did not conduct a meta-analysis. This is a moderate study involving 18 patients [28]. The results showed that there was no significant difference in reductions in the mean diurnal IOP as of 1.5 months of treatment (MD = 0.50 mmHg; $P$ = 0.57). Because only 18 patients were enrolled in this cross-over study, which is fewer than the calculated optimal number, and because this small sample study was commercially funded, it was necessary to downgrade the quality of the study to low due to its risk of bias, imprecision, and publication bias.

**Meta-analyses of mean IOP.** *LTFC vs. TTFC* This meta-analysis included four parallel studies [13, 15, 16, 20] and one cross-over study [23], comprising a total of 652 patients. Aside

**Table 2. IOP characteristics in included studies.**

| First author/ Year/Reference | Treatment | Route | Mean baseline IOP (mm Hg) [mean (SD)] | Type of IOP measurement | Time points (hours after dosing) | End point of measurement (months) | Run-in (weeks) | Wash-out (weeks)[&] |
|---|---|---|---|---|---|---|---|---|
| **Pachimkul 2011** | LTFC vs TTFC | M | 27.1 (8.1)/ 21.1 (6.2) | 24-hour Curve (9) | 1,4,7,10,13,16,19,22,25 | 0.5 | Standard[#] | |
| **Topouzis 2007** | LTFC vs TTFC | M | 25.5 (3.8)/ 26.0 (3.7) | Diurnal Curve (3) | 0,2,7 | Combined 0.5,1.5,3,6,9 | Standard | |
| **Konstas 2014** | LTFC vs TTFC | E | 21.5 (1.6) | 24-hour Curve (5) | 2,6,10,14,18 | 3 | Latanoprost 3 | No |
| **Rigollet 2011** | LTFC vs TTFC vs BiTFC | E | 27.6/28/26.4 | Diurnal single (1) | 12 | 1 | 4 | |
| **Guven Yilmaz 2018** | LTFC vs TTFC vs BiTFC | E | 14.9 (2.5)/14.4 (2.8)/ 13.5(4.2) | 24-hour Curve (6) | 2,6,10,14,18,22 | 1.2 | Latanoprost 4 | |
| **Centofanti 2009** | LTFC vs BiTFC | E | 22.1 (2.7)/ 22.7 (2.1) | Diurnal Curve (3) | 12,14,18 | 1 | No | |
| **Martinez 2007b** | LTFC vs BiTFC | E | 22.2 (0.8)/ 22.3 (0.9) | Diurnal Curve (3) | 12,15,18 | 1 | No | |
| **Martinez 2009** | LTFC vs BiTFC | E | 22.0 (1.0)/ 22.0 (1.0) | Diurnal Curve (7) | 10,12,14,16,18,20,24 | 3 | Timolol 6 | 6 |
| **Cvenkel 2008** | LTFC vs DTFC | M/ M +E | 20.7 (2.3) | Diurnal Curve (7) | 0,2,4,6,8,10,12 | 1.5 | Timolol 6 | No |
| **Eren 2012** | LTFC vs DTFC | E/ M +E | 25.1 (2.8) | 24-hour Curve (6) | 3,7,11,15,19,23 | 1.5 | 6 | 6 |
| **Januleviciene 2009** | LTFC vs DTFC | M/ M +E | 20.6 (3.3)/ 22.1 (2.7) | Diurnal single (1) | Within 12 | 1 | Timolol 4 | |
| **Konstas 2004** | LTFC vs DTFC | M/ M +E | 20.1 (2.0)/ 20.2 (1.9) | Diurnal Curve (7) | 0,2,4,6,8,10,12 | 2 | Timolol 4 | 8 |
| **Konstas 2008** | LTFC vs DTFC | E/ M +E | 22.1 (3.5) | 24-hour Curve (6) | 2,6,10,14,18,22 | 3 | Latanoprost 12 | No |
| **Martinez 2007a** | LTFC vs DTFC | E/ M +E | 26.6 (3.5) | Diurnal single (1) | 12/1 | 1 | 4 | 4 |
| **Miglior 2010** | LTFC vs DTFC | E/ M +E | 26.5 (2.7)/ 27.0 (3.2) | Diurnal Curve (3) | 12,16,22/ 0,4,10 | 3 | Standard | |
| **Shin 2004** | LTFC vs DTFC | M/ M +E | 27.9 (3.6)/ 27.5 (3.1) | Diurnal Curve (3) | 0,4,8 | 3 | Standard | |
| **Hommer 2012** | LTFC vs BTFC | M/ M +E | 25.5 (3.1) | Diurnal Curve (3) | 0,4,8 | 1.5 | Standard | No |

IOP, intraocular pressure; LTFC, 0.005% Latanoprost/0.5% timolol; DTFC, 2.0% Dorzolamide/0.5% timolol; BTFC, 0.2% Brimonidine/0.5% timolol; TTFC, 0.004% Travoprost/0.5% timolol; BiTFC, 0.03% Bimatoprost/0.5% timolol; M, morning; E, evening; POAG, primary open-angle glaucoma; OHT, ocular hypertension.

[#]The washout periods lasted four weeks for b-adrenergic receptor antagonists and prostaglandins, two weeks for adrenergic agonists, and five days for cholinergic agonists and carbonic anhydrase inhibitors; Washout (weeks) was only for cross-over trials.

from the study by Pachimkul (2011), all other studies included IOP results of one month or more. Meta-analysis showed that there was no statistical difference in the mean IOP between the LTFC group and TTFC group (MD: 0.07 mmHg, $P = 0.78$) (Fig 2). There was mild heterogeneity between studies ($I^2 = 29\%$, $P = 0.23$). The sensitivity analysis showed that it originated from the only high risk-of-bias study included in this meta-analysis. However, there was still no significant difference in the mean IOP between the treatment groups (MD: 0.08 mmHg, $P = 0.76$; $I^2 = 0\%$). Therefore, we did not downgrade the quality of this meta-analysis due to inconsistency of results. Four moderate risk-of-bias studies and one high risk-of-bias study were included in this meta-analysis. The risk of bias was mainly from these domains: four studies without blinding of participants and personnel, and four studies with high risk of bias

**Table 3. Risk of bias in the included studies.**

| First author/Year/Reference | Randomization | Allocation concealment | Blinding | | | With a pre-specified sample size | Incomplete outcome data | | Risk of bias |
|---|---|---|---|---|---|---|---|---|---|
| | | | Participants | Investigators | Examiners | | Withdrawn | ITT | |
| Pachimkul 2011 | Unclear | Unclear | Unclear | Unclear | Unclear | No | 20% | No | Moderate |
| Topouzis 2007 | Adequate | Unclear | Yes | Yes | Yes | Yes | 18.60% | No | Moderate |
| Konstas 2014 | Unclear | Unclear | No | No | Yes | Yes | 4.50% | Yes | Moderate |
| Rigollet 2011 | Adequate | Unclear | No | No | Yes | Yes | 9.20% | No | Moderate |
| Guven Yilmaz 2018 | Unclear | Unclear | No | No | Yes | No | 7.40% | Yes [#] | High |
| Centofanti 2009 | Adequate | Unclear | No | Yes | Yes | No | 0% | Yes | Moderate |
| Martinez 2007b | Adequate | Unclear | No | No | Yes | Yes | 0% | Unclear | Moderate |
| Martinez 2009 | Adequate | Unclear | No | No | Yes | Yes | 0% | Yes | Moderate |
| Cvenkel 2008 | Unclear | Unclear | Yes | Yes | Yes | Yes | 0% | Yes | Low |
| Eren 2012 | Unclear | Unclear | Yes | Yes | Yes | Yes | 0% | Unclear | Low |
| Januleviciene 2009 | Unclear | Unclear | Broken blinding | Broken blinding | No | No | 0% | Unclear | Moderate |
| Konstas 2004 | Unclear | Unclear | Yes | Yes | Yes | No | 3% | No | Moderate |
| Konstas 2008 | Adequate | Adequate | Yes | Yes | Yes | No | 8.80% | No | Moderate |
| Martinez 2007a | Adequate | Unclear | No | No | Yes | No | 0% | Yes | Moderate |
| Miglior 2010 | Adequate | Unclear | No | No | Yes | Yes | 4.80% | No | High |
| Shin 2004 | Unclear | Unclear | No | No | Yes | Yes | 3.60% | Yes | Moderate |
| Hommer 2012 | Unclear | Unclear | Yes | Yes | Yes | Yes | 0% | Unclear | Moderate |

[#]Loss to follow-up imbalance between groups. ITT, intention-to-treat analysis.

due to incomplete outcome data. Therefore, we were required to downgrade (−1) the quality of this meta-analysis to moderate due to the study limitations.

*LTFC vs. BiTFC* This meta-analysis included a total of 321 patients from four parallel studies [13, 16, 24, 29] and one cross-study [25], all of which were administered the respective combinations at night and were followed up with for at least one month. A study by Yilmaz et al. (2018) measured the mean IOP for 24 hours, whereas the IOP included in other studies was the mean IOP of the diurnal measurements. In two studies [24, 29], the percent of POAG and OHT patients was less than 50%. Meta-analysis showed that the mean IOP of the end point in the LTFC group was 0.76 mmHg higher than that in the BiTFC group (P < 0.00001), which was consistent across studies ($I^2$ = 0%) (Fig 3). The sensitivity analysis found that the methodological variables and study risk-of-bias variables had no impact on the results. Four

**Table 4. Meta-analyses of mean IOP and IOP fluctuation.**

| Control | No. of Studies | No. of Participants | Mean Difference (95% CI, mmHg) | Test for Overall Effect (P) | Heterogeneity ($I^2$, %) | Test for Heterogeneity (P) |
|---|---|---|---|---|---|---|
| **Mean IOP** | | | | | | |
| TTFC | 5 | 652 | 0.07 [−0.43, 0.58] | 0.78 | 29 | 0.23 |
| BiTFC | 5 | 321 | 0.76 [0.49, 1.04] | < 0.00001 | 0 | 0.6 |
| DTFC | 8 | 841 | −0.31[−0.65, 0.03] | 0.07 | 0 | 0.64 |
| **IOP fluctuation** | | | | | | |
| TTFC | 2 | 148 | 0.13 [−0.73, 0.99] | 0.81 | 19 | 0.29 |
| BiTFC | 2 | 172 | 1.09 [0.62, 1.56] | < 0.00001 | 0 | 0.63 |
| DTFC | 2 | 128 | 0.25 [−0.50, 1.00] | 0.52 | 0 | 0.9 |

LTFC, 0.005% Latanoprost/0.5% timolol; DTFC, 2.0% Dorzolamide/0.5% timolol; TTFC, 0.004% Travoprost/0.5% timolol; BiTFC, 0.03% Bimatoprost/0.5% timolol.

**Table 5. Quality assessment of meta-analyses.**

| Control group | Mean follow time (months) | Limitations | Inconsistency | Indirectness | Imprecision | Publication bias | Quality assessment |
|---|---|---|---|---|---|---|---|
| **Mean IOP** | | | | | | | |
| **TTFC** | 8.2 | Yes[ab] | No | No | No | No | Moderate |
| **BiTFC** | 3.96 | Yes[ab] | No | No | No | No | Moderate |
| **DTFC** | 2 | Yes[acd] | No | No | No | No | Moderate |
| **IOP fluctuation** | | | | | | | |
| **TTFC** | 3 | Yes[bcd] | No | No | No | No | Moderate |
| **BiTFC** | 3 | Yes[abcd] | No | Yes[e] | No | No | Low |
| **DTFC** | 2.25 | Yes[bc] | No | No | No | No | Moderate |

[a]Patients and personnel were not blinded

[b]Loss to follow-up and failure to adhere to the intention-to-treat principle

[c]Unclear randomization

[d]No pre-specified sample size

[e]Half of participants are not POAG or OHT.RCT, randomized controlled trial; LTFC, 0.005% Latanoprost/0.5% timolol; DTFC, 2.0% Dorzolamide/0.5% timolol; TTFC, 0.004% Travoprost/0.5% timolol; BiTFC, 0.03% Bimatoprost/0.5% timolol.

moderate risk-of-bias studies and one high risk-of-bias study were included in this meta-analysis. The risk of bias was mainly from these domains: all five studies were without blinding of participants and personnel, and two studies had high risk of bias due to incomplete outcome data. We downgraded (−1) the quality of this meta-analysis due to study limitations. We did not downgrade the quality of this meta-analysis due to indirectness. Since only 18.3% of the total number of patients included in the analysis were not diagnosed as POAG or OHT, the sensitivity analysis indicated that the indirectness of the population had no impact on the results (test of subgroup difference: *P* = 0.66). We assessed the quality of this meta-analysis as moderate.

*LTFC vs. DTFC* This meta-analysis included three parallel studies [17, 18, 22] and five cross-over studies [14, 19, 21, 26, 27] involving a total of 841 patients, including two large parallel studies involving more than 200 patients. The administration times of DTFC were in the morning and evening, and LTFC was administered in the morning for half of the studies and in the evening for the other half of the studies. Two studies measured and reported the mean IOP over 24 hours, while the rest reported only diurnal IOP. The difference between the two treatments was not statistically significant (MD: −0.31 mmHg, *P* = 0.07), and there was no heterogeneity between these studies (I$^2$ = 0%) (Fig 4). The sensitivity analysis found that the methodological variables and study risk of bias variables had no impact on the results. Two low

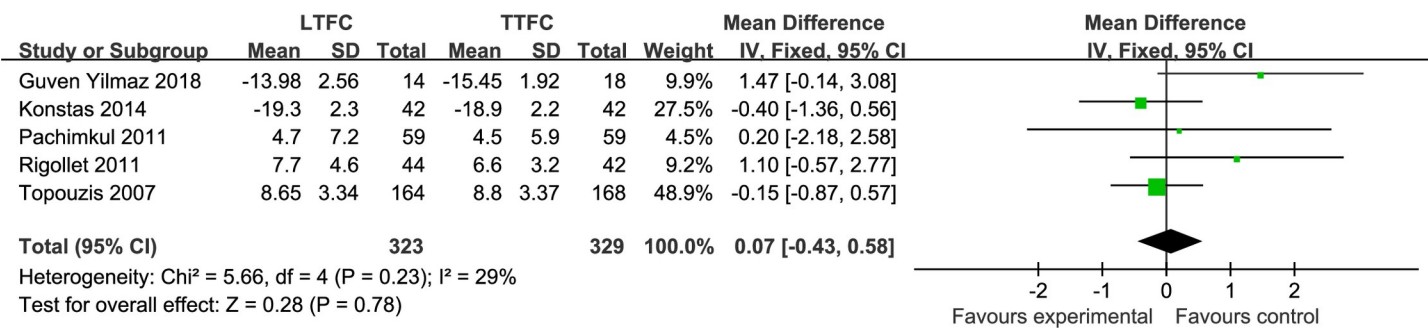

**Fig 2. Forest plot of the mean IOP, comparison of LTFC and TTFC.**

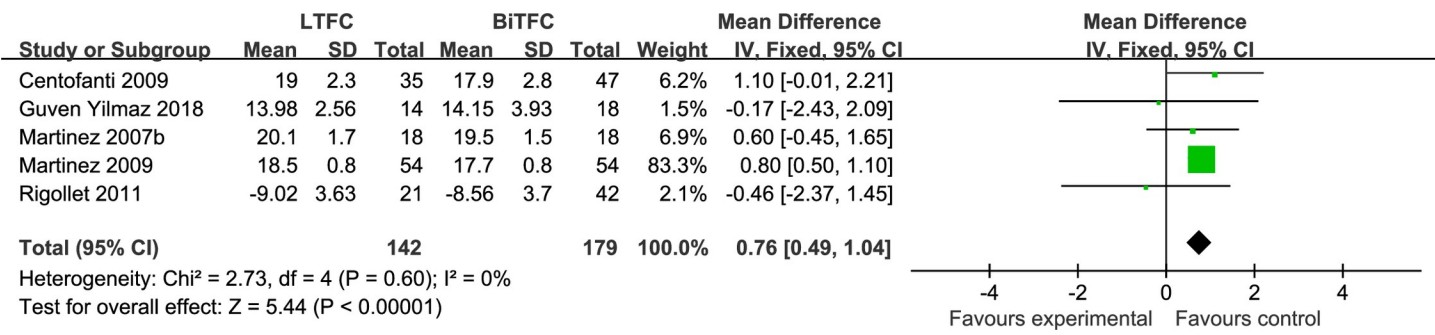

**Fig 3. Forest plot of the mean IOP, comparison of LTFC and BiTFC.**

risk-of-bias studies, five moderate risk-of-bias studies, and one high risk-of-bias study were included in this meta-analysis. The risk of bias was mainly from these domains: five studies did not report the detailed method of randomization, four studies lacked blinding of participants and personnel, and four studies did not report a pre-specified sample size. After the quality was downgraded (-1) due to the limitations of the studies, we assessed the quality of the analysis as moderate.

**Meta-analyses of IOP fluctuation.** *LTFC vs. TTFC* Two studies [13, 23] reported differences in IOP fluctuations at the LTFC and TTFC endpoints; one of these studies was a crossover study of 42 patients reporting 24-hour IOP fluctuations, while the other study was a parallel study of 32 patients reporting both diurnal and nocturnal IOP fluctuations. The analysis showed that there was no significant difference in IOP fluctuations between the LTFC group and the TTFC group (MD = 0.13 mmHg, $P = 0.77$), and the heterogeneity was mild ($I^2 = 19\%$, $P = 0.29$) (Fig 5). Sensitivity analysis showed that the heterogeneity was mainly from the diurnal administration time (test of subgroup difference: $I^2 = 55\%$, $P = 0.13$). However, we did not downgrade the quality of this meta-analysis due to inconsistency of results, as there was still no significant difference in the effect of mean IOP between the treatment groups after removing the diurnal results (MD: 0.03 mmHg, $P = 0.88$; $I^2 = 0\%$, $P = 0.45$). There was uncertainty in the randomization and allocation-concealment methods in the cross-over study; this study was assessed as a moderate risk-of-bias study [23]. In the parallel study, there was uncertainty regarding the randomization method, no pre-specified sample size, and an imbalance among the missing patients; it was assessed as a high risk-of-bias study [13]. Therefore, the meta-analysis quality was downgraded (-1) to moderate due to the limitations of these studies.

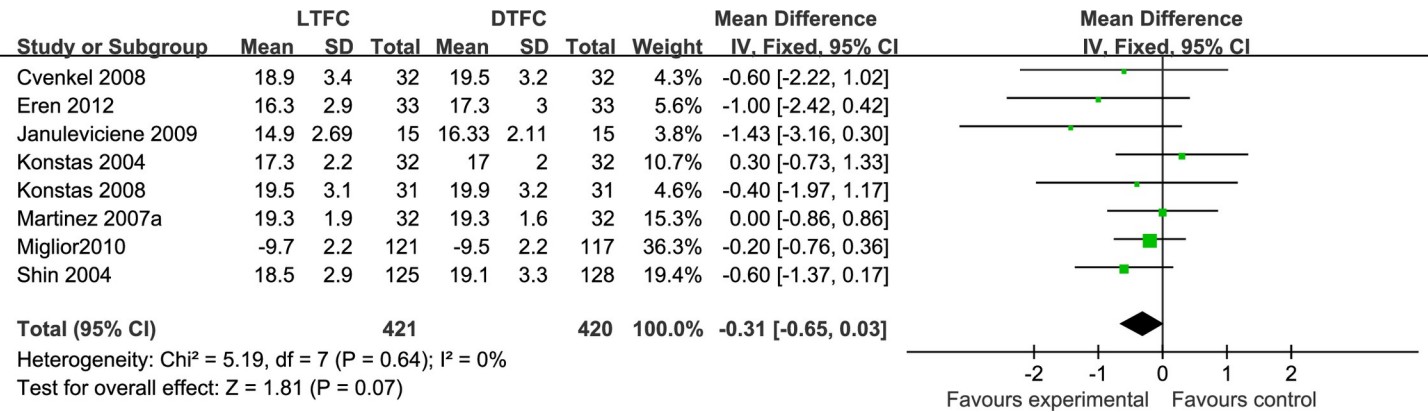

**Fig 4. Forest plot of the mean IOP, comparison of LTFC and DTFC.**

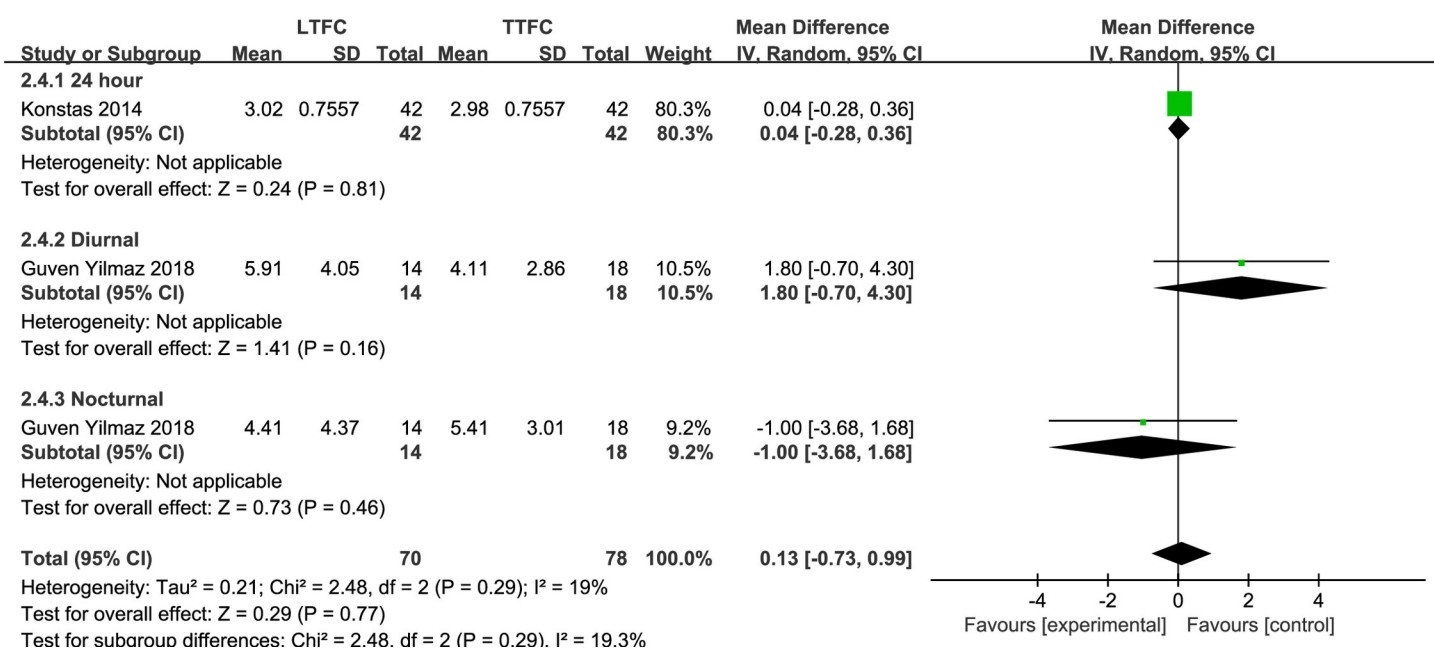

**Fig 5. Forest plot of the IOP fluctuation, comparison of LTFC and TTFC.**

*LTFC vs. BiTFC* A cross-over study [25] of 54 patients reported diurnal IOP fluctuation, and a parallel group study [13] of 32 patients reported diurnal and nocturnal IOP fluctuations. When both drugs were administered once at night, LTFC showed significantly higher IOP fluctuations (MD: 1.09 mmHg, *P* < 0.00001) compared to BiTFC; there was no heterogeneity in this meta-analysis (Fig 6). The parallel study was assessed as a high risk-of-bias study and the cross-over study was assessed as a moderate risk-of-bias study. Overall, 42.8% of the total number of patients included in the analysis were not diagnosed as POAG or OHT. Therefore, we downgraded the quality of the IOP fluctuation meta-analysis to low due to indirectness of target population and study limitations.

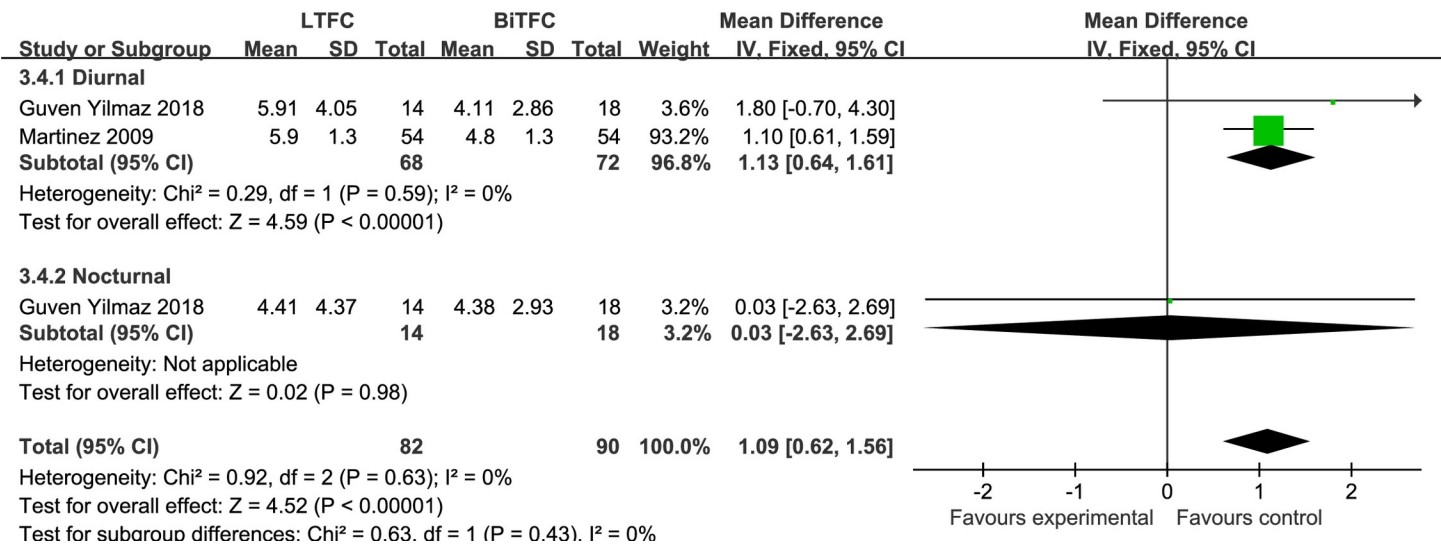

**Fig 6. Forest plot of the IOP fluctuation, comparison of LTFC and BiTFC.**

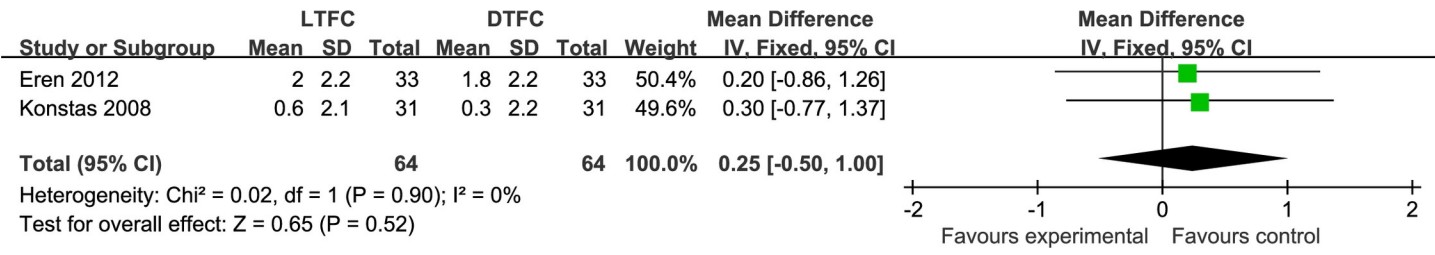

**Fig 7. Forest plot of the IOP fluctuation, comparison of LTFC and DTFC.**

*LTFC vs. DTFC* Two cross-over studies [14, 26] analyzed 24 hours of IOP fluctuations and found no significant differences between the administration of LTFC once a night and DTFC twice a day (MD: 0.25 mmHg, *P* = 0.52) (Fig 7). One study had a low risk of bias and the other had a moderate risk of bias. There were no other domains in which the quality of the evidence was downgraded; thus, we determined that this meta-analysis was of moderate quality.

## Discussion

This systematic review of RCTs showed that LTFC was as effective as TTFC and DTFC but worse than BiTFC in controlling IOP for POAG or OHT patients (Table 4). Additionally, the quality of these analyses was assessed as moderate, except for one low-grade analysis that compared the IOP fluctuation of LTFC and BiTFC (Table 5).

The effectiveness of the fixed-combination therapies was demonstrated by meta-analysis in two parts. First, the mean IOP at one month of the LTFC group was similar to that of the TTFC and DTFC groups but was 0.76 mmHg higher than that of the BiTFC group. Second, we found that there was no significant difference of IOP fluctuation between the LTFC group and the TTFC or DTFC groups at one month; however, the IOP fluctuation was 1.09 mmHg higher in the LTFC group than in the BiTFC group. The quality evaluation of these meta-analyses found that the reason for lowering their quality rating was mainly due to study limitations and indirectness of evidence. Limitations of the studies appeared in all the analyses. The main reasons for study limitations included a lack of participant and personnel blinding, incomplete outcome data, and premature termination of the study. As such, we lowered the rating due to these study limitations. Indirectness of evidence only appeared in the analysis of IOP fluctuation comparing LTFC with BiTFC. Overall, 42.8% of the patients with pseudoexfoliative glaucoma were included in the analysis. A cross-over study of pseudoexfoliative glaucoma patients showed that the IOP fluctuation in the TTFC group was significantly lower than that in the LTFC group (3.4±1.3 mmHg vs. 4.1±1.6 mmHg; *P* < 0.01), and the difference between groups for this parameter was 0.7 mmHg [30]. In contrast, our moderate quality analysis showed that the difference was only 0.13 mmHg in patients with POAG or OHT, suggesting that there would be differences in the effect of the drug on reducing IOP in different patients. Therefore, we lowered the grade of this analysis due to population indirectness.

Our analysis confirmed existing beliefs of the comparative effectiveness of single- or adjunctive-used glaucoma drugs. A network meta-analysis reported the IOP-lowering effect of single glaucoma drugs, which showed that latanoprost could reduce the mean IOP by 4.85 mmHg, travoprost by 4.83 mmHg, and bimatoprost by 5.61 mmHg [31]. Our meta-analysis showed that LTFC is as effective as TTFC but worse than BiTFC in controlling mean IOP. The efficacies of the fixed combinations are consistent with their components. This network meta-analysis also reported that dorzolamide reduced the mean IOP by 2.49 mmHg, which was 1.36 mmHg worse than that of latanoprost. Our meta-analysis also showed that DTFC was only

0.31 mmHg worse than that of LTFC. The meta-analysis studied the effectiveness of concomitant and fixed combinations of glaucoma drugs and found the following: the IOP changes for 2% dorzolamide in concomitant use with 0.5% timolol was 4.9 mmHg, and the IOP changes for the fixed 2% dorzolamide and 0.5% timolol combination was also 4.9 mmHg [32]. Both treatments were used twice daily. The IOP changes for 0.005% latanoprost (used once daily) in concomitant use with 0.5% timolol twice daily was 6 mmHg, and the fixed 0.005% latanoprost and 0.5% timolol combination used once daily was 3.0 mmHg. A direct meta-analysis confirmed this difference [33]. We inferred that the inconsistency of effectiveness between these two fixed combinations and their single agent was mainly related to the change in the frequency of the drug.

There are some meta-analyses [5, 6, 7, 8] that have studied the comparative effectiveness of fixed combinations from different aspects. For example, some meta-analyses [5, 6] studied the relative IOP reduction with indirect comparisons. A systematic review showed that LTFC reduced IOP by 33% and DTFC reduced IOP by 26%; however, only 20 patients were enrolled in the DTFC group in this study [5]. Another meta-analysis [6] showed that IOP reduction was 34.9% for TTFC, 34.3% for BiTFC, 33.9% for LTFC, and 29.9% for DTFC. The results of these analyses are not consistent with the results of our present study, and this discrepancy is likely related to the differing research methods. For example, indirect comparisons have an influence on the baseline, which subsequently affects the results.

Other meta-analyses have studied the absolute IOP reduction with direct comparisons. One meta-analysis studied the comparative effectiveness of the fixed combination of prostaglandin and timolol [7]; this meta-analysis with four studies showed that LTFC and TTFC had a comparative effectiveness in diurnal IOP (MD = 0.08 mmHg), but the results had obvious heterogeneity ($I^2 = 89\%$, $P < 0.00001$). The sensitivity analysis showed that the heterogeneity may reflect differences in sample size, study types, interventions, baseline treatment, study duration, and many other factors. Our meta-analysis included five clinical trials that were consistent with the conclusions of this meta-analysis; specifically, that LTFC and TTFC had a comparative effectiveness in diurnal IOP (MD = 0.07 mmHg). Our meta-analysis had no obvious heterogeneity ($I^2 = 29\%$, $P = 0.23$), and the sensitivity analysis, according to research quality, found that the heterogeneity resulted from a poor-quality trial. The difference in heterogeneity between this meta-analysis and our analysis might be associated with the target population. This systematic review included studies with normal-tension glaucoma or pseudoexfoliative glaucoma patients, but we excluded these studies. Another meta-analysis involving four clinical trials found that LTFC was significantly worse than BiTFC in reducing the diurnal mean IOP (MD = 0.88 mmHg) [8]. We note that there was significant heterogeneity in this analysis ($I^2 = 66\%$, $P = 0.03$), and sensitivity analysis showed that the study center (single or multiple) may have been the main contributor to this heterogeneity. Our meta-analysis of five included studies showed that the mean IOP at the end point of LTFC was 0.76 mmHg higher than that of BiTFC, and there was no heterogeneity ($I^2 = 0\%$). The difference of the heterogeneity was related to the outcomes. We selected the IOP of the end point as the outcome index instead of the change of IOP from baseline to avoid errors caused by the numerical transformation. Another previous meta-analysis included seven studies that reported the same mean diurnal IOP-lowering effect between LTFC and DTFC (MD 0.16 mmHg, $P = 0.51$), but heterogeneity was found in this analysis ($I^2 = 34.9\%$, $P = 0.61$) [34]. Our present results are consistent with these findings; however, there was no heterogeneity in our analysis ($I^2 = 0\%$). The sensitivity analysis also showed that methodological variables and study risk-of-bias variables had no impact on the results.

There were several limitations in our system review in terms of the included participants, control group, and outcome measurements. First, our target population was POAG and OHT

patients, and therefore we excluded some studies that only included normal-tension glaucoma and pseudoexfoliative glaucoma patients. In a previous study, we found that there were differences in the efficacy of the drug to reduce IOP in different patients [30]. The defined target population would reduce the heterogeneity but also limit the results in terms of the application to other patients. Second, our meta-analysis did not include all the different fixed combinations. An open-label study reported the difference between a tafluprost/timolol fixed combination and LTFC, with 131 patients who were followed up for three months [12]. This study found that the IOP before and 4–6 hours after administration were not significantly different. However, there was no blinding in this study and the sample size was smaller (n = 60 vs. n = 55) than originally planned (75 patients in each group) without explanation; thus, we downgraded the quality to moderate due to these study limitations. Finally, we selected the IOP at one month after treatment as the outcome. Although the defined measurement time points limit the application of research results to other shorter or longer time points, the heterogeneity of the measurement results can be reduced. This time point was ample for the drug to lower IOP, while the number of patients lost to follow-up was most likely small.

Although our study had some similarities to those of previous meta-analyses, as well as some limitations as described above, this work analyzed IOP fluctuations, did have a smaller heterogeneity, and evaluated the quality of the meta-analyses according to the GRADE system for the fixed combination of anti-glaucoma drugs.

## Conclusions

In summary, LTFC was as effective as TTFC and DTFC but less ideal than BiTFC for controlling IOP for POAG or OHT patients. Additionally, the quality of these meta-analyses was moderate, except for one low-grade analysis that compared the IOP fluctuation of LTFC and BiTFC. Additional research with a low risk of bias and large sample size is needed to evaluate the IOP fluctuation lowering effect of FCs.

## Supporting information

**S1 File. PRISMA checklist.**
(DOC)

**S2 File. Search strategy.**
(DOC)

## Author Contributions

**Conceptualization:** Yi Xing, Shaohua Huang.

**Data curation:** Yi Xing.

**Formal analysis:** Yi Xing.

**Investigation:** Yi Xing, Lijuan Zhu, Ke Zhang.

**Methodology:** Yi Xing.

**Project administration:** Yi Xing, Lijuan Zhu, Ke Zhang.

**Software:** Yi Xing, Lijuan Zhu, Ke Zhang.

**Supervision:** Yi Xing.

**Validation:** Shaohua Huang.

**Visualization:** Yi Xing.

**Writing – original draft:** Yi Xing.

**Writing – review & editing:** Shaohua Huang.

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
