## [Decision Letter · Decision Letter 0]

21 Jan 2020

PONE-D-19-26644

The efficacy of the fixed combination of latanoprost and timolol versus other fixed combinations for primary open-angle glaucoma and ocular hypertension: a systematic review and meta-analysis

PLOS ONE

Dear Author,

Thank you for submitting your manuscript to PLOS ONE. After careful consideration, we feel that it has merit but does not fully meet PLOS ONE’s publication criteria as it currently stands. Therefore, we invite you to submit a revised version of the manuscript that addresses the points raised during the review process.

We would appreciate receiving your revised manuscript by February 10. To enhance the reproducibility of your results, we recommend that if applicable you deposit your laboratory protocols in protocols.io, where a protocol can be assigned its own identifier (DOI) such that it can be cited independently in the future. For instructions see: http://journals.plos.org/plosone/s/submission-guidelines#loc-laboratory-protocols

We look forward to receiving your revised manuscript.

Kind regards,

Simone Donati, M.D.

Academic Editor

PLOS ONE

Journal Requirements:

2. Please move your figures from the supporting information files to the main body of the manuscript. Note there are no length restrictions for papers published in PLOS ONE.

Reviewers' comments:

Reviewer's Responses to Questions

**Comments to the Author**

1. Is the manuscript technically sound, and do the data support the conclusions?

Reviewer #1: Yes

Reviewer #2: Yes

2. Has the statistical analysis been performed appropriately and rigorously? 

Reviewer #1: Yes

Reviewer #2: Yes

3. Have the authors made all data underlying the findings in their manuscript fully available?

Reviewer #1: Yes

Reviewer #2: Yes

4. Is the manuscript presented in an intelligible fashion and written in standard English?

Reviewer #1: Yes

Reviewer #2: Yes

5. Review Comments to the Author

Reviewer #1: This is a well-done study in which the authors obtained interesting results, even if some of them were already known. This metaanalysis outlined the efficacy of fixed combination.

There are some English tense to correct, but the manuscript is well written.

Reviewer #2: Line 83. There are more fixed combination (brinzolamide/timolol, brimonidine/brinzolamide and the oldest pilocarpine/timolol). Why did you forget all these?

Line 384. Please change DTFC according with other abbreviations. Really, I would prefer all the abbreviations as DTFC than FCDT…

Line 442. Please avoid “To the best of our knowledge” and “first”. You can’t read all the published Literature and all the databases.

6. PLOS authors have the option to publish the peer review history of their article (what does this mean?). If published, this will include your full peer review and any attached files.

Reviewer #1: No

Reviewer #2: No

---

## [Author Response · Author response to Decision Letter 0]

31 Jan 2020

We have studied comments carefully, and made revision according to the reviewers’ suggestion. Revised portion are marked in red in the revised manuscript. The responses to the reviewers’ comments are as following:

Reviewer #1:

This is a well-done study in which the authors obtained interesting results, even if some of them were already known. This metaanalysis outlined the efficacy of fixed combination.

There are some English tense to correct, but the manuscript is well written.

Response: We have checked and corrected English tense errors in the manuscript.

Reviewer #2:

1.Line 83. There are more fixed combination (brinzolamide/timolol, brimonidine/brinzolamide and the oldest pilocarpine/timolol). Why did you forget all these?

Response: Special thanks for your good comments. We have added these fixed combinations in the introduction. We have reported that only four types of fixed combinations were found in the control groups of the included studies in the systematic review.

2.Line 384. Please change DTFC according with other abbreviations. Really, I would prefer all the abbreviations as DTFC than FCDT…

Response: We have accepted your suggestion and have modified the abbreviations of all fixed combinations. 

3.Line 442. Please avoid “To the best of our knowledge” and “first”. You can’t read all the published Literature and all the databases.

Response: We have modified this section according to the recommendations.

We hope that the revised manuscript is now acceptable for publication in your journal.

With kind regards.

---

## [Editor Report · Decision Letter 1]

12 Feb 2020

The efficacy of the fixed combination of latanoprost and timolol versus other fixed combinations for primary open-angle glaucoma and ocular hypertension: a systematic review and meta-analysis

PONE-D-19-26644R1

Dear Authors,

We are pleased to inform you that your manuscript has been judged scientifically suitable for publication and will be formally accepted for publication once it complies with all outstanding technical requirements.

With kind regards,

Simone Donati, M.D.

Academic Editor

PLOS ONE

Additional Editor Comments (optional):

Dear Authors,

thank you for your revision.

The manuscript is now accepted for publication.

Best regards
---

## [Editor Report · Acceptance letter]

14 Feb 2020

PONE-D-19-26644R1 

The efficacy of the fixed combination of latanoprost and timolol versus other fixed combinations for primary open-angle glaucoma and ocular hypertension: a systematic review and meta-analysis 

Dear Dr. Huang:

I am pleased to inform you that your manuscript has been deemed suitable for publication in PLOS ONE. Congratulations! Your manuscript is now with our production department. 

With kind regards,

on behalf of

Dr. Simone Donati 

Academic Editor

PLOS ONE